

# Evaluation of IWV from the numerical weather prediction WRF model with PPP GNSS processing for Bulgaria

Tzvetan Simeonov[1,a], Dmitry Sidorov[2,b], Felix Norman Teferle[2], Georgi Milev[3], and Guergana Guerova[1]

[1]Faculty of Physics, Sofia University "St. Kliment Ohridski", Bulgaria

[a]now at GeoForschungsZentrum (GFZ) Potsdam, Germany

[2]Université du Luxembourg, Luxembourg

[b]now at Astronomical Institute University of Bern, Switzerland

[3]Space Research and Technology Center - Bulgarian Academy of Sciences, Bulgaria

*Correspondence to:* Tzvetan Simeonov (simeonov@phys.uni-sofia.bg)

**Abstract.** Global Navigation Satellite Systems (GNSS) meteorology is an established operational service providing hourly updated GNSS tropospheric products to the National Meteorologic Services (NMS) in Europe. In the last decade through the ground-based GNSS network densification and new processing strategies like Precise Point Positioning (PPP) it has become possible to obtain sub-hourly tropospheric products for monitoring severe weather events. In this work one year (January - December 2013) of sub-hourly GNSS tropospheric products (Zenith Total Delay) are computed using the PPP strategy for seven stations in Bulgaria. In order to take advantage of the sub-hourly GNSS data to derive Integrated Water Vapour (IWV) surface pressure and temperature with similar temporal resolution is required. As the surface observations are on 3 hourly basis the first step is to compare the surface pressure and temperature from numerical weather prediction model Weather Forecasting and Research (WRF) with observations at three synoptic stations in Bulgaria. The mean difference between the two data-sets for 1) surface pressure is less than 0.5 hPa and the correlation is over 0.989, 2) temperature the largest mean difference is 1.1° C and the correlation coefficient is over 0.957 and 3) IWV mean difference is in range 0.1-1.1 mm. The evaluation of WRF on annual bases shows IWV underestimation between 0.5 and 1.5 mm at five stations and overestimation at Varna and Rozhen. Varna and Rozhen have also much smaller correlation 0.9 and 0.76. The study of the monthly IWV variation shows that at those locations the GNSS IWV has unexpected drop in April and March respectively. The reason for this drop is likely problems with station raw data. At the remaining 5 stations a very good agreement between GNSS and WRF is observed with high correlation during the cold part of 2013 i.e. March, October and December (0.95) and low correlation during the warm part of 2013 i.e. April to August (below 0.9). The diurnal cycle of the WRF model shows a dry bias in the range of 0.5-1.5 mm. Between 00 and 01 UTC the GNSS IWV tends to be underestimate IWV which is likely due to the processing window used.





The precipitation efficiency from GNSS and WRF show very good agreement on monthly bases with a maximum in May-June and minimum in August-September. The annual precipitation efficiency in 2013 at Lovech and Burgas is about 6 %.

## 1 Introduction

The atmospheric water vapour is a key element of the hydrological cycle and participates in precipitation formation, energy transfer and atmospheric stability. Water vapour has a relatively short lifetime in the atmosphere, from one week to ten days and its complex life cycle includes vertical and horizontal transport, mixing, condensation, precipitation and evaporation. Due to its high temporal and spatial variability atmospheric water vapour is very demanding to observe.

An established method for monitoring water vapour is the radiosonde. In Europe, the radiosonde network consists of 93 stations operated by the National Meteorological Services (NMS) under the EUMETNET-EUCOS project (euc, 2016). The radiosonde provide high vertical resolution data but due to its high cost is operated only one or two times per day at 00 UTC, at 12 UTC or 00 and 12 UTC, respectively. To monitor the high temporal and spatial water vapour variability a new method was developed in the early 1990s using the Global Positioning System (GPS) signal delay. The method was called "GPS meteorology" but with the development of other GNSS, for example Glonass and Galileo, was renamed to "GNSS meteorology". A memorandum of understanding between E-GVAP (the EUMETSAT GNSS water vapour program) and EUPOS (the European Position determination System), which opens opportunities to use GNSS data for Bulgaria and South East Europe is in action since 2012.

Multi technique comparisons (Ning et al., 2012; Buehler et al., 2012; Van Malderen et al., 2014) have demonstrated that GNSS meteorology derived Integrated Water Vapour (IWV) has a root mean square error in the range of 0.4-0.6 mm. A number of studies compare IWV from GNSS and Numerical Weather Prediction (NWP) models in Europe. A recent study by Keernik et al. (2014) found that the HIRLAM NWP model underestimates the IWV by 59 % for values below 12 mm, and overestimates by 6-10 % for values over 25 mm. A study of the COSMO model diurnal IWV cycle over Germany (Tomassini et al., 2002), reports a systematic IWV underestimation larger than 1 mm in the model analysis between 06 and 18 UTC. For Switzerland, Guerova et al. (2003) report a good agreement between model analysis and GNSS in winter but in summer, a significant underestimation of IWV was found in the model, which is well correlated with significant overestimation of light precipitation. For both Germany and Switzerland a systematic underestimation of the diurnal IWV cycle between 6 and 21 UTC in both the model analysis and forecast is reported in Guerova and Tomassini (September 2003).

A recent development in GNSS processing is use of the Precise Point Positioning (PPP) strategy (Zumberge et al., 1997). In contrast to the Precise Network Positioning (PNP) strategy, PPP uses original data without differencing. Since 2013, the International GNSS Service (IGS, (Dow et al., 2009; Caissy et al., 2012)) provides ultra-fast or real-time precise satellite orbit and clock corrections in support of PPP processing (Douša and Vaclavovic, 2014; Li et al., 2014; Yuan et al., 2014; Ahmed et al., 2014). The PPP strategy has the advantage of being computationally much more efficient than PNP and hence can provide estimates for large networks of stations with high temporal resolution (every 5 min). This cannot be achieved by the more conventional PNP strategy and sufficinet IT infrastructure. These new generation GNSS products are yet to be fully





compared with the state-of-the-art NWP models and are of particular interest for very short range weather forecasting of severe weather events (nowcasting).

During the last 5 years GNSS meteorology was developed in Bulgaria within a Marie Curie funded project. As apart of this project a regional database for Bulgaria and Southeast Europe the Sofia University Atmospheric Data Archive (SUADA,

Guerova et al. (2014)) was developed to facilitate the use of GNSS tropospheric products (see Sect. 2.2). In this study for the first time a PPP GNSS processing strategy is applied to seven stations in Bulgaria and GNSS IWV is derived using the Weather Forecasting System (WRF) NWP model. The aim of this work is to evaluate the IWV from GNSS and the WRF NWP model for Bulgaria in 2013. The objectives are to: (1) derive GNSS IWV for Bulgaria using surface pressure and temperature from synop observations and the WRF model, (2) evaluate the WRF model IWV a) on annual and monthly basis and b) the diurnal

cycle and (3) evaluate the precipitation efficiency of the WRF model for Bulgaria in 2013. In Section 2 are presented the GNSS processing strategy and the WRF model set up as well as the data-sets used. Section 3 presents the IWV annual and monthly comparison as well as the diurnal IWV cycle in GNSS and WRF. The summary and conclusions are given in Section 4.

## 2   Data sets and method

### 2.1   Numerical weather prediction model WRF set up

The WRF model is developed in the USA by a collaboration of groups at National Center for Atmospheric Research (NCAR), Mesoscale and Microscale Meteorology Division, the National Oceanic and Atmospheric Administration (NOAA), National Center for Environmental Prediction (NCEP), Earth System Research Laboratory (ESRL), Department of Defence Air Force Weather Agency (AFWA), Naval Research Laboratory (NRL), Center for Analysis and Prediction of Storms (CAPS), Federal Aviation Administration (FAA) and the University of Oklahoma. In this work the WRF v3.4.1 (NCAR, 2016) is computed

for a domain covering Bulgaria with a horizontal resolution of 9 km and a vertical resolution of 44 levels. The following parametrizations schemes for the model physics are selected: 1) Unified Noah land-surface model for the land surface (Chen et al., 1996), 2) Yonsei University (YSU) scheme for the planetary boundary layer (Hong et al., 2006), 3) WRF Single moment Microphysics (WSM) 6-class graupel scheme for the microphysics (Hong and Lim, 2006) and 4) Rapid Radiative Transfer Model (RRTM) for the long/short-wave radiation (Mlawer et al., 1997).

The WRF model output is integrated into the SUADA. Two types of WRF model parameters are archived in SUADA, namely surface parameters (pressure and temperature) and profiles (pressure, temperature, water vapour mixing ratio and the model level height). Presented in Fig. 1 is the SUADA data flow. The surface parameters from WRF are archived in the NWP_IN_1D SUADA table and used to compute the GNSS IWV (Sect. 2.2). The profiles from WRF are archived in the NWP_IN_3D SUADA table and are used to compute the water vapour density at each model level ($\rho_{wv}(z)$) and then by integration over the

model levels the WRF-IWV is obtained as below:

$$IWV = \frac{1}{\rho_w} \int\limits_z^{z_n} \rho_{wv}(z)\, \mathrm{d}z \qquad (1)$$



where $\rho_w$ is density of liquid water, $n$ is the number of model levels.

## 2.2 GNSS processing strategy and tropospheric products

Archived in SUADA are GNSS tropospheric products like Zenith Total Delay (ZTD over 12 000 000 individual observations) and derivatives like IWV (over 55 000) from five GNSS processing strategies and 37 stations in Bulgaria/Southeast Europe for

the period 1997-2013. The temporal resolution of the GNSS data is from 5 minutes to 6 hours.

In this work we use GNSS tropospheric products from the BULgarian intelegent POsitioning System (BULiPOS, http://www.bulipos.eu/ (2016)) GNSS network in Bulgaria. The BULiPOS network of reference stations was established in 2008 and has 26 stations mainly used for navigation and geodetic applications. BULIPOS provided Receiver Independent Exchange Format (RINEX) GNSS data for seven station for 2013 (Fig. 2). The GNSS tropospheric products (Zenith Total Delay, ZTD) were computed

with the NAvigation Package for Earth Observation Satellites (NAPEOS, http://www.positim.com/napeos.html (2016)) software. NAPEOS is developed and maintained by the European Space Operations Centre (ESOC) of the European Space Agency (ESA). NAPEOS is used at ESOC since January 2008. The NAPEOS version 3.3.1 was used for the processing in this study. The processing was performed at the University of Luxembourg using the GMF (Global Mapping Function) (Boehm et al., 2006) and 10° elevation cut-off angle. The data were processed using the PPP strategy employing IGS satellite orbits and

clocks. The computed ZTDs are with a temporal resolution of 300 s (5 min).

The ZTD data is archived in the GNSS_IN SUADA table (Fig. 1). To derive GNSS-IWV the WRF model surface pressure ($p_s$, [hPa]) and temperature ($t_s$, [K]) are used in Eq. 2 (Davis et al., 1985) and Eq. 3,4 (Bevis et al., 1992)

$$ZWD = ZTD - 0.0022768 \frac{p_s}{1 - 0.00266 cos(2\theta) - 0.00028h} \tag{2}$$

$$IWV = \frac{10^6}{(k_3/T_m + k_2')R_v} ZWD, \tag{3}$$

$$T_m = 70.2 + 0.72 t_s, \tag{4}$$

where $k_2' = (17 \pm 10)[\text{K*hPa}^{-1}]$, $k_3 = (3.776 \pm 0.004) * 10^5 [\text{K}^2\text{*hPa}^{-1}]$ are constants derived first by Thayer (1974) and

$R_v = 461.51[\text{J*kg}^{-1}\text{K}^{-1}]$ is the gas constant for water vapour, $T_m$ [K] is the weighted mean atmospheric temperature, $h$ [km] is the height and $\theta$ is the latitude variation of the gravitational acceleration.

The pressure at the GNSS station altitude is calculated using the model pressure at the nearest model grid point. The pressure difference between the GNSS station altitude and the nearest NWP model grid point is calculated using the polytropic





barometric formula Sissenwine et al. (1962):

$$P_g = P_m \left( \frac{T}{T - L(H_g - H_m)} \right)^{\left( \frac{g_0 M_0}{R*L} \right)} \tag{5}$$

where $P_g$ is the pressure at the GNSS station altitude, $P_m$ is the pressure at meteorological station altitude, $T$ [K] is the temperature in meteorological station, $L = 6.5\,K/km$ is tropospheric lapse rate, $H_m$ [km] is the altitude of the meteorological

station, $H_g$ [km] is the altitude of the GNSS station, $g_0 = 9.806\,\frac{m}{s^2}$ is the gravitational acceleration, $M_0 = 28.9644\,\frac{g}{mol}$ is the molar mass of air and $R = 8.31432\,\frac{Nm}{(molK)}$ is the universal gas constant.

## 2.3 Surface observations

Archived in SUADA are also surface observations of: 1) pressure, 2) 2 m temperature and 3) precipitation (PP). The measurements are from the surface observation network (SYNOP) of the National Institute of Meteorology and Hydrology (NIMH)

in Bulgaria and are collected manually every 3 hours (00, 03, 06, 09, 12, 15, 18 and 21 UTC). The data is available from the OGIMET weather information server (ogi, 2016). The surface pressure and temperature are used for derivation of IWV from the GNSS tropospheric products as described in Sect. 2.2. Using surface observations the IWV is derived every 3 hours and is referred to as IWV* in Table 1.

## 2.4 Precipitation efficiency

In order to study water availability Tuller (1971) proposed Precipitation Efficiency (PE) expressed as percentage of the IWV that is converted and measured as precipitation. Bordi et al. (2015) proposed to use GNSS IWV to compute PE. In this work the daily PE is computed as following:

$$PE = \frac{PP}{IWV} \cdot 100 \tag{6}$$

where PP and IWV are daily averaged precipitation and IWV at the station. Below PE computed from GNSS IWV and

observed PP is entitled GNSS PE and from WRF IWV and PP is entitled WRF PE.

## 3 Evaluation of the WRF model IWV with GNSS-IWV for 2013

### 3.1 Evaluation of WRF model surface pressure and temperature

The annual mean, standard deviation and correlation between the surface pressure and temperature from WRF and SYNOP are presented in Table 1. The WRF pressure and temperature are extracted with the temporal resolution of the SYNOP i.e. every 3

25   hours. The correlation coefficient between the two data sets for atmospheric pressure for station Lovech (LOVE) is 0.99 with the mean difference 0.5 hPa. The correlation coefficient for the temperature is 0.96. The largest differences in the two data





sets are observed for December 2013. For station Varna (VARN) the correlation coefficient for the pressure is 1 and for the temperature 0.96 with a mean difference between the SYNOP and NWP-WRF of 0.2 hPa for the pressure and 0.2° C for the temperature. For station Burgas (BURG) the correlation coefficient for the pressure is 1 and for the temperature 0.96. The mean difference for the pressure is 0.1 hPa and 0.2° C for the temperature. The NWP-WRF surface pressure shows an agreement of

0.5 hPa or better with the SYNOP data-set. This allows to take advantage of deriving IWV with the temporal resolution of the GNSS tropospheric products. A comparison between IWV and IWV* for station Burgas is seen in Fig. 3.

### 3.2    WRF-GNSS IWV: annual and monthly mean

A comparison between the GNSS and WRF IWV is presented in Table 3. For Burgas, Lovech, Montana, Shumen and Stara Zagora the correlation coefficient is very high and lies between 0.95 and 0.96. The mean IWV difference is between 0.5 and

1.8 mm. The smallest mean difference is obtained for Shumen and Burgas and is a consequence of the small altitude difference between GNSS station and WRF model height (Table 2). The altitude difference for station Lovech is 107 m and there the largest mean difference of 1.8 mm is obtained. For Varna and Rozhen the correlation coefficient is 0.9 and 0.76, and the mean IWV difference is negative with -0.9 and -3.2 mm respectively.

     The comparison between the GNSS and WRF monthly mean IWV for 2013 is presented in Fig. 4. At all stations with

exception of Rozhen the monthly mean IWV minimum is 10 mm in December 2013 and the maximum is 25 mm in in June 2013. The GNSS and WRF IWV for station Burgas is shown in Fig. 4a. It can be seen that there is good agreement between the monthly mean IWV from GNSS and WRF. The correlation coefficient varies between 0.96 and 0.84. The maximum and minimum correlation is seen in winter and autumn, and spring and summer, respectively. Between stations Shumen (Fig. 4b) and Stara Zagora (Fig. 4c) similarities in the IWV can be observed. The values indicate again a maximum in June and a

minimum in December. For Shumen the lowest correlation is observed in April and it stays low during the spring months. For Stara Zagora the correlation coefficient stays low in with minimum from April till August. Montana (Fig. 4d) is in Northwest Bulgaria where the influence of the Balkan mountains is significant and the interaction with synoptic flows plays a major role for the IWV distribution. The lowest GNSS and WRF IWV values are seen for December 12 mm and the highest for June with 27 mm. For Varna (Fig. 4e) of interest is the difference between GNSS and WRF, which is seen during the months April and

May. From January to April the IWV in the WRF is lower than the GNSS and from May to December it is the opposite. Similar GNSS IWV jump between April and May is seen at Rozhen (Fig. 4f). A possible reason for these changes is the GNSS station set up which needs further investigation.

### 3.3    WRF-GNSS IWV: diurnal cycle

In Fig. 5 half hourly IWV from GNSS and WRF are averaged and plotted for each station. The diurnal cycle of IWV for Burgas

is presented in Fig. 5a. The WRF IWV is between 0.5 and 1.0 mm lower than for GNSS. The mean difference between the two data sets is around 0.5 mm up to 10 UTC. Between 10 and 20 UTC the difference is larger at around 1 mm. At Lovech (Fig. 5b) the difference between GNSS and WRF IWV is between 1.0 and 1.5 mm. This however is expected and is due to the discussed altitude difference (107 m) between the WRF grid point and the GNSS station. It is to be noted that the GNSS, WRF





altitude difference is under 40 m for the other 4 stations. For Montana, Shumen and Stara Zagora WRF has a dry bias relative to the GNSS. For Montana (Fig. 5c) the estimated difference is between 1.2 and 1.7 mm. Larger differences between datasets are seen in the afternoon after 13 UTC (top plots in Fig. 5c). The mean difference in the diurnal IWV variation at Shumen (Fig. 5d) and Stara Zagora (Fig. 5e) between GNSS IWV and WRF IWV is in the range of 0.5 - 1.0 mm. At all stations between 00

and 01 UTC the GNSS has a tendency to underestimate IWV which is likely related to the limits of the beginning and the end of the GNSS processing, called processing window. In the beginning of each processing the GNSS solution is unstable due to lack of initial conditions. The PPP processing uses daily IGS orbits files with jumps in the orbits on the day boundaries. These jumps influence the IWV values.

### 3.4  WRF-GNSS comparison: precipitation efficiency in 2013

Figure 6 shonws the monthly mean precipitation efficiency computed with GNSS and WRF for Burgas and Lovech. The very good agreement between the model and GNSS is apparent at both locations. At Burgas (Fig. 6a) the PE has minimum in August less than 1 % and maximum in May 14 %. For Lovech (Fig. 6b) the maximum PE is in May but is slightly smaller than in Burgas (12 %) and the minimum is in September of about 1 %. The PE at the two stations also shows differences with are expected as the two stations are located in different climatic regions in Bulgaria. While Burgas is in south-east Bulgaria

close to the Black Sea, Lovech is in north-west Bulgaria. The atmospheric circulation in Bulgaria is dependent on the Balkan mountains in middle of the country i.e. south of the range the Mediterranean cyclones are the main source of precipitation, while the north of the range their influence is largely reduced. The annual PE in both stations is in the range of 5.5-5.8 % from GNSS and 5.9-6.0 % from WRF, which is in agreement with the range of 5-10 % for the region found in Tuller (1971).

### 4  Conclusions

In this work GNSS tropospheric products (ZTD) temporal resolution 5 min. are derived using the PPP processing for one year period. In order to take advantage of the high temporal resolution of GNSS products for derivation of IWV the surface pressure and temperature from the NWP WRF model is used. The WRF surface pressure and temperature was evaluated against surface observations from three synoptic stations in Bulgaria. The mean difference for surface pressure between the two data-sets is less than 0.5 hPa and the correlation is over 0.989. For the temperature the largest mean difference is 1.1° C and the correlation

coefficient is over 0.957. The IWV computed with this two data-sets has a mean difference is in range of 0.1-1.1 mm.

The evaluation of WRF on annual bases shows IWV underestimation between 0.5 and 1.5 mm at five stations and overestimation at Varna and Rozhen. Varna and Rozhen have also much smaller correlation 0.9 and 0.76. The study of the monthly IWV variation shows that at those locations the GNSS IWV has unexpected drop in April and March, respectfully. The reason for this drop is likely problems with station raw data. At the remaining 5 stations a very good agreement between the GNSS

and WRF is observed with highest correlation in cold part of the year i.e. March, October and December (over 0.95) and lowest correlation during the warm part of the year i.e. April to August (below 0.9). The diurnal cycle of the WRF model shows a dry





bias in the range 0.5-1.5 mm. Between 00 and 01 UTC the GNSS IWV tends to be underestimated IWV, which is likely due to the processing time window used.

The precipitation efficiency from GNSS and WRF show very good agreement on monthly bases with a maximum in May-June and minimum in August-September. The annual precipitation efficiency in 2013 at Lovech and Burgas is about 6 %.

This work is a first step of setting up GNSS Analysis Centre for tropospheric products at Sofia University (Sofia University GNSS Analysis Centre - SUGAC). SUGAC is a collaboration between Sofia University and Space Research and Technology Institute of the Bulgarian Academy of Sciences. The consistent PPP processing of 7 GNSS station demonstrates a potential to provide IWV with high temporal resolution for validation of the state-of-the-art NWP models. However, operational provision of GNSS tropospheric products is delayed due to the availability of real time RINEX data. The team at Sofia University is

working towards establishment of dedicate GNSS network for atmospheric remote sensing products in Bulgaria.

*Acknowledgements.* Tzvetan Simeonov acknowledge the support provided by COST - European Cooperation in Science and Technology project "Advanced Global Navigation Satellite Systems tropospheric products for monitoring severe weather events and climate" (GNSS4SWEC) for Short-Term Scientific Mission (STSM) to University of Luxembourg. This work is supported by a Marie Curie International Reintegration Grant (FP7-PEOPLE-2010-RG) within the 7th European Community Framework Programme and Sofia University

grant. The technical support by Dr. Stoyan Pisov and Dr. Elisaveta Peneva is greatly appreciated.



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

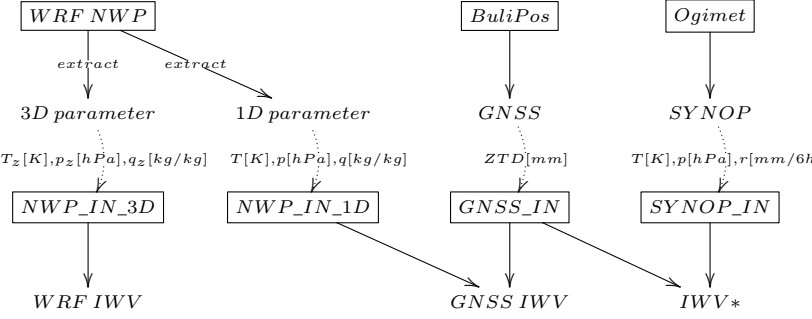

**Figure 1.** SUADA data flow.



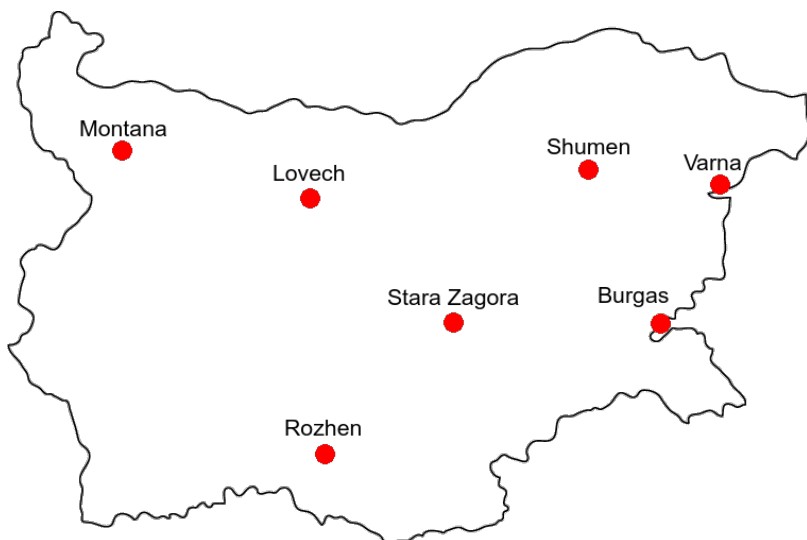

**Figure 2.** Map of the ground based stations of the Bulipos GNSS network. The red markers show the station locations.

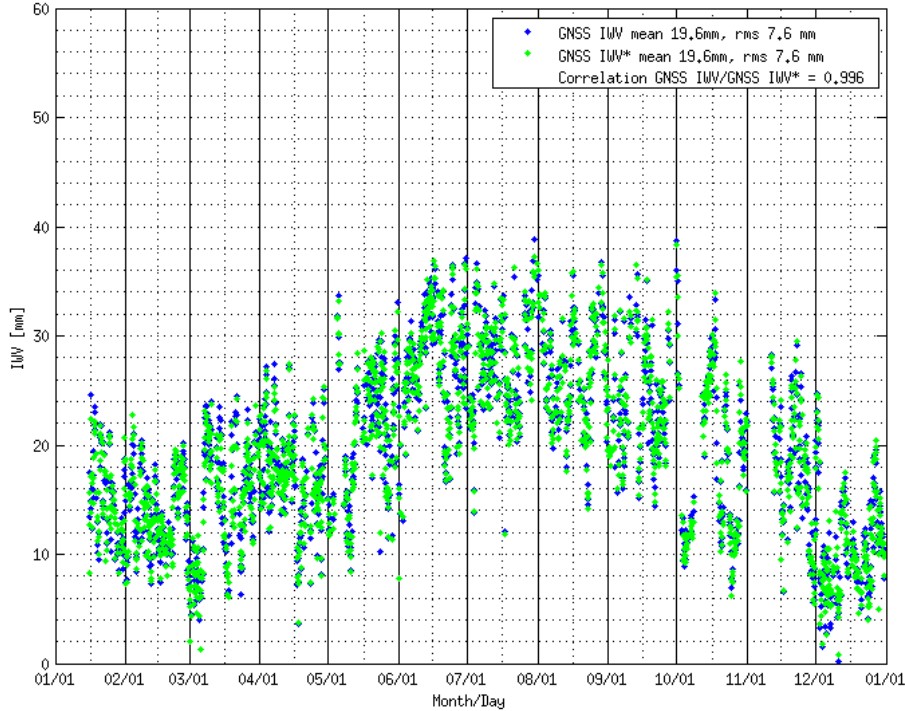

**Figure 3.** GNSS IWV and IWV* for station Burgas in 2013.





**Figure 4.** Top plots: IWV monthly mean from GNSS and WRF for: a) Burgas, b) Shumen, c) Stara Zagora, d) Montana, e) Varna and f) Rozhen in 2013. Bottom plots: correlation coefficient.





**Figure 5.** Top plots: IWV diurnal cycle from GNSS and WRF for: a) Burgas, b) Lovech, c) Montana, d) Shumen and e) Stara Zagora. Bottom plots: IWV difference GNSS minus WRF.



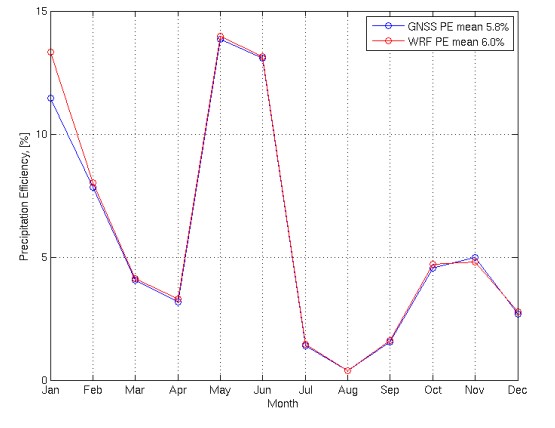
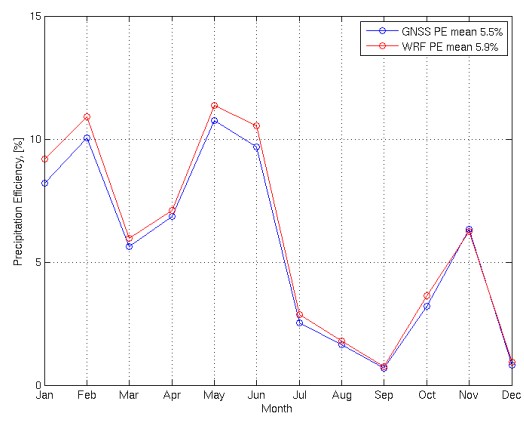

a)  b)

**Figure 6.** Precipitation efficiency from GNSS and WRF for: a) Burgas and b) Lovech.

| Station | Pressure WRF mean | Pressure SYNOP mean | Pressure WRF STD | Pressure SYNOP STD | Correlation coefficient |
|---|---|---|---|---|---|
| Lovech | 1015.4 | 1015.9 | 7.4 | 7.4 | 0.989 |
| Burgas | 1015.6 | 1015.8 | 7.3 | 7.4 | 0.995 |
| Varna | 1015.6 | 1015.8 | 7.2 | 7.4 | 0.996 |
| Station | Temperature WRF mean | Temperature SYNOP mean | Temperature WRF STD | Temperature SYNOP STD | Correlation coefficient |
| Lovech | 15.1 | 14.0 | 8.7 | 9.6 | 0.957 |
| Burgas | 13.9 | 14.2 | 8.3 | 8.3 | 0.960 |
| Varna | 13.7 | 13.9 | 8.1 | 8.3 | 0.975 |
| Station | GNSS IWV mean | IWV* mean | GNSS IWV STD | IWV* STD | Correlation coefficient |
| Lovech | 18.2 | 17.0 | 7.6 | 7.3 | 0.999 |
| Burgas | 19.6 | 19.6 | 7.6 | 7.6 | 0.996 |
| Varna | 17.4 | 17.3 | 6.9 | 7.0 | 0.999 |

**Table 1.** Surface pressure (top) and temperature (middle) and IWV (bottom) from SYNOP and WRF. Mean (column 2 and 3), standard deviation (column 4 and 5 ) and correlation coefficient (column 6).

| Station | GNSS elevation [m] | WRF level elevation [m] | SYNOP elevation [m] | GNSS-WRF diff [m] |
|---|---|---|---|---|
| Burgas | 71 | 34 | 28 | +66 |
| Stara Zagora | 227 | 254 | - | -27 |
| Shumen | 268 | 243 | - | +25 |
| Montana | 203 | 225 | - | -22 |
| Lovech | 243 | 350 | 221 | -107 |
| Varna | 62 | 96 | 43 | -34 |
| Rozhen | 1779 | 1431 | - | +348 |

**Table 2.** GNSS and meteorological stations and WRF surface level elevations



| Station | GNSS IWV mean | GNSS IWV STD | WRF IWV mean | WRF IWV STD | GNSS/WRF IWV correlation | GNSS-WRF IWV mean difference |
|---|---|---|---|---|---|---|
| Montana | 19.4 | 7.6 | 17.9 | 7.3 | 0.95 | 1.5 |
| Lovech | 18.2 | 7.6 | 16.4 | 7.2 | 0.96 | 1.8 |
| Shumen | 18.0 | 7.5 | 17.5 | 7.5 | 0.96 | 0.5 |
| Burgas | 19.6 | 7.6 | 19.0 | 7.6 | 0.96 | 0.6 |
| Stara Zagora | 18.5 | 7.5 | 17.3 | 7.4 | 0.96 | 1.2 |
| Varna | 17.4 | 6.9 | 18.3 | 7.9 | 0.90 | -0.9 |
| Rozhen | 7.9 | 4.2 | 11.1 | 5.3 | 0.76 | -3.2 |

**Table 3.** Mean (column 2 and 4), standard deviation (STD, column 3 and 5), correlation coefficient (column 6) and mean difference (column 7) of GNSS and WRF IWV for 2013.