# Peer review of "Evaluation of IWV from the numerical weather prediction WRF model with PPP GNSS processing for Bulgaria"

_Atmospheric Measurement Techniques, 2016_

## Referee Comment (RC1) · Anonymous Referee #1 · 11 Aug 2016

Dear authors,

I have read your manuscript and, unfortunately, I cannot support its publication in AMT journal. A brief justification of my decision to reject the manuscript is given in the following.

Although the subject of the manuscript is of scientific interest, the approach followed is quite shallow. In particular, the analysis of results does not go in depth and refrains to simply presenting statistics, either in tables or figures. A discussion on the causes of the computed differences is almost absent, while there is no discussion relevant to the available literature. Further, the added-value of the presented work is not very obvious. A simple verification exercise is certainly important, especially for new "products", but

is it enough for supporting a publication?

---

## Referee Comment (RC2) · Anonymous Referee #2 · 7 Sep 2016

The paper contains interesting comparisons of data.

The new idea and proposal of new method or methodology have not been articulated enough AMT journal's requirement. The chapter "methodology" should be separated and included new products.

The description of GNSS processing strategy is too detailed and refers to a known standard processing of GNSS data in PPP mode.

The paper requires significant changes for showing the integration model not only comparisons of data from different sources.

The WRF model needs reference: Skamarock WC, Klemp JB, Dudhia J, Gill DO,

[Figure]

Barker DM, Duda MG, Huang XY, Wang W, Powers JG (2008) A Description of the Advanced Research WRF Version 3. NCAR Tech. Note NCAR/TN-475+STR, doi:10.5065/D68S4MVH;

The same ideas integration of GNSS processing data and WRF model outputs are presented in the paper: Wilgan K., Hurter F., Geiger A., Rohm W., Bosy J. (2016) Tropospheric refractivity and zenith path delays from least-squares collocation of meteorological and GNSS data. Journal of Geodesy, DOI: 10.1007/s00190-016-0942-5, URL: http://link.springer.com/article/10.1007/s00190-016-0942-5

The conclusions should show more progress than confirm already known from literature results.

I cannot accept this publication in this form in AMT journal and major revision is required.

---

## Author Comment (AC1) · 18 Nov 2016

Dear reviewer,

Thank you for the time invested in helping us improve the presentation of our work! Please find below our response to the your recommendations underlined.

1) Although the subject of the manuscript is of scientific interest, the approach followed is quite shallow. In particular, the analysis of results does not go in depth and refrains to simply presenting statistics, either in tables or figures.

This work is a first step in application of GNSS for atmospheric remote sensing in Bulgaria in particular for validation of the NWP model WRF. We agree that NWP model

evaluation with GNSS has been performed in other regions but for Bulgaria such studies are not available. The dominant features of the atmospheric circulation in Bulgaria and South-east Europe are the Mediterranean cyclones which have complex interaction with topography. Thus it is mandatory to evaluate the model performance. Our experience with simulation of intense summer precipitation has shown that the WRF model has high sensitivity to the convective parameterisation for example and this was the reason to perform an yearly check of the performance of the selected set-up. A detrending using annual and semiannual components was done for all stations. Stations Varna and Rozhen have inhomogenities in the datasets, so the detrending for those stations is highly dependent on the jumps. Stations Stara Zagora, Burgas and Montana have gaps in the beginning or the end of the year, so the estimated trends are not representative. Only stations Lovech and Shumen have long enough datasets for annual trend analysis, so the data for them is included into the manuscript.

The following paragraph is added to section 3.2 with supporting figure 7: The datasets of Lovech and Shumen have the shortest gaps among the studied stations. These two stations were detrended using the following annual fitting function, as proposed by Ning (2012) : $y = at + b\cos(2\pi t) + c\sin(2\pi t) + d\cos(4\pi t) + e\sin(4\pi t)$ where b and c are the annual coefficients and d and e - the semi-annual, while a is a linear trend component. These coefficients were determined using least-square analysis. The correlation between the datasets is high (0.913 for Lovech and 0.901 for Shumen) after subtracting the seasonal variation (Fig. 7). This analysis could not be performed for the other 5 stations, because of the gaps in the datasets, which influence the trend analysis of both the annual variation and the monthly change in IWV.

2) A discussion on the causes of the computed differences is almost absent, while there is no discussion relevant to the available literature.

The following paragraph is added to section 3.2: Further work was carried out to investigate the possible reasons for reported drop in GNSS-IWV values at station Varna and Rozhen. The manual investigation of the raw GNSS data showed that at station Varna

wrong antenna model as reported in the raw data. After the antenna model correction the processing resulted to an IWV increase by 2 mm in December 2013. For station Rozhen the manual investigation did not show any mistakes thus the problem remains there.

The following paragraph is added to section 3.3: The WRF model has an under-estimation of diurnal IWV cycle at all stations in the range of 0.5-1.5 mm Guerova and Tomassini (September 2003) report a systematic underestimation of the diurnal IWV cycle between 6 and 21 UTC in COSMO analysis and forecast for Germany and Switzerland. It is not possible to link our study with the one done with COSMO model as each NWP model has its own characteristics (Guerova et al., 2016). NWP models are set up differently, and have different performance, depending on selected region, resolution, season and parametrisation schemes. Our experience with simulation of intense summer precipitation in Bulgaria during 2012 has shown that the WRF model has high sensitivity to the convective parameterisation scheme used and this prompted the present study to a full year check of the the model performance of the selected set-up.

3) A simple verification exercise is certainly important, especially for new "products", but is it enough for supporting a publication?

The GNSS tropospheric products derived from PPP have the advantage of providing high temporal and spatial resolution, which is in high demand for short range weather forecasting and nowcasting application. PPP is a new product for atmospheric analysis using the GNSS Meteorology method. This is one of the first studies, using PPP processing in high temporal resolution, intending to observe IWV variations.

With kind regards, Tzvetan Simeonov on behalf of co-authors

Please also note the supplement to this comment:
http://www.atmos-meas-tech-discuss.net/amt-2016-152/amt-2016-152-AC1-

supplement.pdf

[Figure]

**Supplement:**

[revised manuscript text omitted]
 The Precipitation Efficiency (PE) is a value, calculated for each station, which is descriptive for air masses in two time scales: Cloud Microphysics Precipitation Efficiency (CMPE) (Braham Jr, 1952) and Large-Scale Precipitation Efficiency (LSPE) Tuller (1971) . In this study we are assessing LSPE, which is representative for the

region climate and climate variations. PE is expressed as percentage of the IWV that is converted and measured as precipitation. Bordi et al. (2015) proposed to use GNSS IWV to compute PE. In this work the daily PE is computed as following:

$$PE = \frac{PP}{IWV} \cdot 100 \tag{6}$$

where PP and IWV are daily averaged precipitation and IWV at the station.  Precipitation efficiency gives a long-term
5  indication of stability of the atmosphere. In this study the two analysed stations for PE are in low altitudes; PE computed from GNSS IWV and observed PP is entitled GNSS PE and from WRF IWV and PP is entitled WRF PE.

**3 Evaluation of the WRF model IWV with GNSS-IWV for 2013**

**3.1 Evaluation of WRF model surface pressure and temperature**

The annual mean, standard deviation and correlation between the surface pressure and temperature from WRF and SYNOP are
10  presented in Table 1. The WRF pressure and temperature are extracted with the temporal resolution of the SYNOP i.e. every 3 hours. The correlation coefficient between the two data sets for atmospheric pressure for station Lovech (LOVE) is 0.99 with the mean difference 0.5 hPa. The correlation coefficient for the temperature is 0.96. The largest differences in the two data sets are observed for December 2013. For station Varna (VARN) the correlation coefficient for the pressure is 1 and for the temperature 0.96 with a mean difference between the SYNOP and NWP-WRF of 0.2 hPa for the pressure and 0.2° C for the
15  temperature. For station Burgas (BURG) the correlation coefficient for the pressure is 1 and for the temperature 0.96. The mean difference for the pressure is 0.1 hPa and 0.2° C for the temperature. The NWP-WRF surface pressure shows an agreement of 0.5 hPa or better with the SYNOP data-set. This allows to take advantage of deriving IWV with the temporal resolution of the GNSS tropospheric products. A comparison between IWV and IWV* for station Burgas is seen in Fig. 3.

**3.2 WRF-GNSS IWV: annual and monthly mean**

20  A comparison between the GNSS and WRF IWV is presented in Table 3. For Burgas, Lovech, Montana, Shumen and Stara Zagora the correlation coefficient is very high and lies between 0.95 and 0.96. The mean IWV difference is between 0.5 and 1.8 mm. The smallest mean difference is obtained for Shumen and Burgas and is a consequence of the small altitude difference between GNSS station and WRF model height (Table 2). The altitude difference for station Lovech is 107 m and there the largest mean difference of 1.8 mm is obtained. For Varna and Rozhen the correlation coefficient is 0.9 and 0.76, and the mean
25  IWV difference is negative with -0.9 and -3.2 mm respectively.

The comparison between the GNSS and WRF monthly mean IWV for 2013 is presented in Fig. 4. At all stations with exception of Rozhen the monthly mean IWV minimum is 10 mm in December 2013 and the maximum is 25 mm in in June 2013. The GNSS and WRF IWV for station Burgas is shown in Fig. 4a. It can be seen that there is good agreement between the monthly mean IWV from GNSS and WRF. The correlation coefficient varies between 0.96 and 0.84. The maximum and

minimum correlation is seen in winter and autumn, and spring and summer, respectively. Between stations Shumen (Fig. 4b) and Stara Zagora (Fig. 4c) similarities in the IWV can be observed. The values indicate again a maximum in June and a minimum in December. For Shumen the lowest correlation is observed in April and it stays low during the spring months. For Stara Zagora the correlation coefficient stays low in with minimum from April till August. Montana (Fig. 4d) is in Northwest

5 Bulgaria where the influence of the Balkan mountains is significant and the interaction with synoptic flows plays a major role for the IWV distribution. The lowest GNSS and WRF IWV values are seen for December 12 mm and the highest for June with 27 mm. For Varna (Fig. 4e) of interest is the difference between GNSS and WRF, which is seen during the months April and May. From January to April the IWV in the WRF is lower than the GNSS and from May to December it is the opposite. Similar GNSS IWV jump between April and May is seen at Rozhen (Fig. 4f). A possible reason for these changes is the GNSS station

10 set up which needs further investigation.

Further work was carried out to investigate the possible reasons for reported drop in GNSS-IWV values at station Varna and Rozhen. The manual investigation of the raw GNSS data showed that at station Varna wrong antenna model as reported in the raw data. After the antenna model correction the processing resulted to an IWV increase by 2 mm in December 2013. For station Rozhen the manual investigation did not show any mistakes thus the problem remains there.

15 The datasets of Lovech and Shumen have the shortest gaps among the studied stations. These two stations were detrended using the following annual fitting function, as proposed by Ning (2012) :

$$y = at + b\cos(2\pi t) + c\sin(2\pi t) + d\cos(4\pi t) + e\sin(4\pi t) \tag{7}$$

where $b$ and $c$ are the annual coefficients and $d$ and $e$ - the semi-annual, while $a$ is a linear trend component. These coefficients were determined using least-square analysis. The correlation between the datasets is high (0.913 for Lovech and 0.901 for

20 Shumen) after subtracting the seasonal variation (Fig. 7). This analysis could not be performed for the other 5 stations, because of the gaps in the datasets, which influence the trend analysis of both the annual variation and the monthly change in IWV.

**3.3 WRF-GNSS IWV: diurnal cycle**

[revised manuscript text omitted]

The diurnal IWV cycle is investigated for Bulgaria for 2013. The diurnal variations of atmospheric water vapour affect long wave radiation, absorption of solar radiation and is related to processes such as atmospheric stability, diurnal variation of moist convection and precipitation, surface wind convergence and evapotranspiration. Thus it is important to evaluate the IWV cycle of the WRF model shows for Bulgaria. At all stations the models has a dry bias in the range 0.5-1.5 mm. Between 00 and 01 UTC the GNSS IWV tends to be underestimated IWV , which is likely due to the processing time window used. Studies with other models show a link between IWV underestimation and overestimation of light precipitation. Such study could not be performed for Bulgaria as the precipitation observations are only available as accumulated 6 hourly values.

In order to link the IWV and precipitation the precipitation efficiency coefficient is computed at two stations. Precipitation efficiency gives the percentage of IWV converted in precipitation. The precipitation efficiency from GNSS and WRF show very good agreement on monthly bases with a maximum in May-June and minimum in August-September. The annual precipitation efficiency in 2013 at Lovech and Burgas is about 6 %, which is within the typical values range for low elevation stations in moderate and continental climates. It will be interesting to investigate the precipitation efficiency at the mountainous stations but co-location of GNSS and reliable surface observations is a limiting factor for such analysis.

[revised manuscript text omitted]

---

## Author Comment (AC2) · 18 Nov 2016

Dear reviewer,

Thank you for the time invested in helping us improve the presentation of our work! Please find below our response to the your recommendations (underlined text).

1.) The paper contains interesting comparisons of data. The new idea and proposal of new method or methodology have not been articulated enough AMT journal's requirement. The chapter "methodology" should be separated and included new products.

The new product, discussed in this paper is not the method, but the implementation of a combination of known methods for a new geographical area. However, we note that

this is maybe not sufficiently well articulated. Thus in the introduction is included the following text: Use of GNSS derived Water Vapour (WV) in Europe is a well established techniques but there exist a large difference on regional level (Guerova et al., 2016). While in West and central Europe the topic has reached maturity in South and particularly South-east Europe it is currently under development. Regarding the derivation of IWV we use NWP atmospheric parameters in combination with the GNSS tropospheric products, which is not widely used. Thus we include a validation of the NWP model parameters (surface pressure and temperature) in order to evaluate their accuracy and precision. In addition, a use of PPP derived tropospheric products is gaining interest among the atmospheric community, as it provides high temporal and spatial resolution for nowcasting applications (intense precipitation, hail and thunderstorms). The PPP products can be also used for evaluation of the NWP models. To the best of our knowledge, this is a first attempt to use them for NWP evaluation. Computed is also the Precipitation Efficiency, which reflects the water availability in the model and the atmosphere and even more importantly combines two components of the hydrologic cycle the IWV and precipitation.

2) The description of GNSS processing strategy is too detailed and refers to a known standard processing of GNSS data in PPP mode.

In our opinion, the PPP processing part of section 2 is balanced and gives the necessary details so thus the processing strategy is sufficiently well documented. As the Napeos software is not widely used for derivation of tropospheric products we took extra care to provide the processing parameters used.

3) The paper requires significant changes for showing the integration model not only comparisons of data from different sources.

This manuscript does not attempt to develop integration model as the number of GNSS stations is very limited (only 7) and sparsely distributed. In addition, the processing was conducted for limited time (2 weeks) which restricted the possible outcome. Evaluation

of WRF model with GNSS-IWV was not done for South-east Europe. In South-east Europe the hydrologic regime is driven by Mediterranean cyclones and evaluation of model performance is further challenged by the topography of the region. This study is unlikely to be repeated in near real future or turned into a real/near real-time operational service due to lack of assessable real time data from the GNSS stations in Bulgaria.

4.) The WRF model needs reference: Skamarock WC, Klemp JB, Dudhia J, Gill DO, Barker DM, Duda MG, Huang XY, Wang W, Powers JG (2008) A Description of the Advanced Research WRF Version 3. NCAR Tech. Note NCAR/TN-475+STR, doi:10.5065/D68S4MVH;

Thank you this suggestion! The reference is included in the manuscript:

In this work the WRF v3.4.1 (NCAR, 2016; Skamarock et al, 2008) computed is computed for a domain covering Bulgaria with a horizontal resolution of 9 km and a vertical resolution of 44 levels.

5.) The same ideas integration of GNSS processing data and WRF model outputs are presented in the paper: Wilgan K., Hurter F., Geiger A., Rohm W., Bosy J. (2016) Tropospheric refractivity and zenith path delays from least-squares collocation of meteorological and GNSS data. Journal of Geodesy, DOI: 10.1007/s00190-016-0942-5, URL: http://link.springer.com/article/10.1007/s00190-016-0942-5

Thank you pointing to the paper by Wilgan et al. (2016)! The paper is recently published and we had not had the chance to study it before preparing the manuscript. In Wilgan et al. (2016) the refractivity of the atmosphere is calculated from the available meteorological and model data in order to compare ZTD values from GNSS and from the WRF model. In our study we are focused on comparing the IWV data, obtained from both the WRF model and the GNSS/WRF as we only have sparsely distributed stations as explained in our reply to question 3. However, Wilgan et al. (2016) is added in the introduction of the revised manuscript: For Poland, Wilgan et al. (2016) developed an integration model for estimating ZTD using WRF and reports good agreement

between the ZTD estimation of WRF, compared to both GNSS and radiosonde measurements.

6) The conclusions should show more progress than confirm already known from literature results. I cannot accept this publication in this form in AMT journal and major revision is required.

In the conclusion section of the revised manuscript is included new paragraph as below: The diurnal IWV cycle is investigated for Bulgaria for 2013. The diurnal variations of atmospheric water vapour affect long wave radiation, absorption of solar radiation and is related to processes such as atmospheric stability, diurnal variation of moist convection and precipitation, surface wind convergence and evapotranspiration. Thus it is important to evaluate the IWV cycle of the WRF model for Bulgaria. At all stations the models has a dry bias in the range 0.5-1.5 mm. Studies with other models show a link between IWV underestimation and overestimation of light precipitation. Such study could not be performed for Bulgaria as the precipitation observations are only available as accumulated 6 hourly values.

In order to link the IWV and precipitation the precipitation efficiency coefficient is computed at two stations. Precipitation efficiency gives the percentage of IWV converted in precipitation. The precipitation efficiency from GNSS and WRF show very good agreement on monthly bases with a maximum in May-June and minimum in August-September. The annual precipitation efficiency in 2013 at Lovech and Burgas is about 6 %, which is within the typical values range for low elevation stations in moderate and continental climates. It will be interesting to investigate the precipitation efficiency at the mountainous stations but co-location of GNSS and reliable surface observations is a limiting factor for such analysis.

Between 00 and 01 UTC the GNSS IWV tends to be underestimated, which is likely due to the processing time window used. In the beginning of each processing the GNSS solution is unstable due to lack of initial conditions. The PPP processing uses daily

IGS orbits files with jumps in the orbits on the day boundaries. These jumps influence the IWV values.

With kind regards, Tzvetan Simeonov on behalf of co-authors

Please also note the supplement to this comment:
http://www.atmos-meas-tech-discuss.net/amt-2016-152/amt-2016-152-AC2-supplement.pdf

[Figure]

**Supplement:**

**Evaluation of IWV from the numerical weather prediction WRF model with PPP GNSS processing for Bulgaria**

Tzvetan Simeonov1,a, Dmitry Sidorov2,b, Felix Norman Teferle2, Georgi Milev3, and Guergana Guerova1

1Faculty of Physics, Sofia University "St. Kliment Ohridski", Bulgaria

anow at GeoForschungsZentrum (GFZ) Potsdam, Germany

2Université du Luxembourg, Luxembourg

bnow at Astronomical Institute University of Bern, Switzerland

3Space Research and Technology Center - Bulgarian Academy of Sciences, Bulgaria

Correspondence to: Tzvetan Simeonov (simeonov@phys.uni-sofia.bg)

[revised manuscript text omitted]

 station, which is descriptive for air masses in two time scales: Cloud Microphysics Precipitation Efficiency (CMPE) (Braham Jr, 1952) and Large-Scale Precipitation Efficiency (LSPE) Tuller (1971) . In this study we are assessing LSPE, which is representative for the

region climate and climate variations. PE is expressed as percentage of the IWV that is converted and measured as precipitation. Bordi et al. (2015) proposed to use GNSS IWV to compute PE. In this work the daily PE is computed as following:

$$PE = \frac{PP}{IWV} .100 \tag{6}$$

where PP and IWV are daily averaged precipitation and IWV at the station. Below-Precipitation efficiency gives a long-term
indication of stability of the atmosphere. In this study the two analysed stations for PE are in low altitudes; PE computed from
GNSS IWV and observed PP is entitled GNSS PE and from WRF IWV and PP is entitled WRF PE.

**3 Evaluation of the WRF model IWV with GNSS-IWV for 2013**

**3.1 Evaluation of WRF model surface pressure and temperature**

- The annual mean, standard deviation and correlation between the surface pressure and temperature from WRF and SYNOP are
  presented in Table 1. The WRF pressure and temperature are extracted with the temporal resolution of the SYNOP i.e. every 3 hours. The correlation coefficient between the two data sets for atmospheric pressure for station Lovech (LOVE) is 0.99 with the mean difference 0.5 hPa. The correlation coefficient for the temperature is 0.96. The largest differences in the two data sets are observed for December 2013. For station Varna (VARN) the correlation coefficient for the pressure is 1 and for the temperature 0.96 with a mean difference between the SYNOP and NWP-WRF of 0.2 hPa for the pressure and 0.2° C for the temperature. The NWP-WRF surface pressure shows an agreement of
- 0.5 hPa or better with the SYNOP data-set. This allows to take advantage of deriving IWV with the temporal resolution of the GNSS tropospheric products. A comparison between IWV and IWV\* for station Burgas is seen in Fig. 3.

**3.2 WRF-GNSS IWV: annual and monthly mean**

20 A comparison between the GNSS and WRF IWV is presented in Table 3. For Burgas, Lovech, Montana, Shumen and Stara Zagora the correlation coefficient is very high and lies between 0.95 and 0.96. The mean IWV difference is between 0.5 and 1.8 mm. The smallest mean difference is obtained for Shumen and Burgas and is a consequence of the small altitude difference between GNSS station and WRF model height (Table 2). The altitude difference for station Lovech is 107 m and there the largest mean difference of 1.8 mm is obtained. For Varna and Rozhen the correlation coefficient is 0.9 and 0.76, and the mean

25 IWV difference is negative with -0.9 and -3.2 mm respectively.

The comparison between the GNSS and WRF monthly mean IWV for 2013 is presented in Fig. 4. At all stations with exception of Rozhen the monthly mean IWV minimum is 10 mm in December 2013 and the maximum is 25 mm in in June 2013. The GNSS and WRF IWV for station Burgas is shown in Fig. 4a. It can be seen that there is good agreement between the monthly mean IWV from GNSS and WRF. The correlation coefficient varies between 0.96 and 0.84. The maximum and

minimum correlation is seen in winter and autumn, and spring and summer, respectively. Between stations Shumen (Fig. 4b) and Stara Zagora (Fig. 4c) similarities in the IWV can be observed. The values indicate again a maximum in June and a minimum in December. For Shumen the lowest correlation is observed in April and it stays low during the spring months. For Stara Zagora the correlation coefficient stays low in with minimum from April till August. Montana (Fig. 4d) is in Northwest

- 5 Bulgaria where the influence of the Balkan mountains is significant and the interaction with synoptic flows plays a major role for the IWV distribution. The lowest GNSS and WRF IWV values are seen for December 12 mm and the highest for June with 27 mm. For Varna (Fig. 4e) of interest is the difference between GNSS and WRF, which is seen during the months April and May. From January to April the IWV in the WRF is lower than the GNSS and from May to December it is the opposite. Similar GNSS IWV jump between April and May is seen at Rozhen (Fig. 4f). A possible reason for these changes is the GNSS station
- 10 set up which needs further investigation.

Further work was carried out to investigate the possible reasons for reported drop in GNSS-IWV values at station Varna and Rozhen. The manual investigation of the raw GNSS data showed that at station Varna wrong antenna model as reported in the raw data. After the antenna model correction the processing resulted to an IWV increase by 2 mm in December 2013. For station Rozhen the manual investigation did not show any mistakes thus the problem remains there.

15 The datasets of Lovech and Shumen have the shortest gaps among the studied stations. These two stations were detrended using the following annual fitting function, as proposed by Ning (2012) :

$$\underbrace{y = at + b\cos(2\pi t) + c\sin(2\pi t) + d\cos(4\pi t) + e\sin(4\pi t)}_{\text{(2)}}$$

where b and c are the annual coefficients and d and e - the semi-annual, while a is a linear trend component. These coefficients were determined using least-square analysis. The correlation between the datasets is high (0.913 for Lovech and 0.901 for

(7)

20 Shumen) after subtracting the seasonal variation (Fig. 7). This analysis could not be performed for the other 5 stations, because of the gaps in the datasets, which influence the trend analysis of both the annual variation and the monthly change in IWV.

**3.3 WRF-GNSS IWV: diurnal cycle**

25

In Fig. 5 half hourly IWV from GNSS and WRF are averaged and plotted for each station. The diurnal cycle of IWV for Burgas is presented in Fig. 5a. The WRF IWV is between 0.5 and 1.0 mm lower than for GNSS. The mean difference between the two data sets is around 0.5 mm up to 10 UTC. Between 10 and 20 UTC the difference is larger at around 1 mm. At Lovech

- (Fig. 5b) the difference between GNSS and WRF IWV is between 1.0 and 1.5 mm. This however is expected and is due to the discussed altitude difference (107 m) between the WRF grid point and the GNSS station. It is to be noted that the GNSS, WRF altitude difference is under 40 m for the other 4 stations. For Montana, Shumen and Stara Zagora WRF has a dry bias relative to the GNSS. For Montana (Fig. 5c) the estimated difference is between 1.2 and 1.7 mm. Larger differences between datasets
- 30 are seen in the afternoon after 13 UTC (top plots in Fig. 5c). The mean difference in the diurnal IWV variation at Shumen (Fig. 5d) and Stara Zagora (Fig. 5e) between GNSS IWV and WRF IWV is in the range of 0.5 1.0 mm. At all stations between 00 and 01 UTC the GNSS has a tendency to underestimate IWV which is likely related to the limits of the beginning and the end

of the GNSS processing, called processing window. In the beginning of each processing the GNSS solution is unstable due to lack of initial conditions. The PPP processing uses daily IGS orbits files with jumps in the orbits on the day boundaries. These jumps influence the IWV values.

The WRF model has an underestimation of diurnal IWV cycle at all stations in the range of 0.5-1.5 mm Guerova and Tomassini (Septemb

- a systematic underestimation of the diurnal IWV cycle between 6 and 21 UTC in COSMO analysis and forecast for Germany 5 and Switzerland. It is not possible to link our study with the one done with COSMO model as each NWP model has its own characteristics (Guerova et al., 2016). NWP models are set up differently, and have different performance, depending on selected region, resolution, season and parametrisation schemes. Our experience with simulation of intense summer precipitation in Bulgaria during 2012 has shown that the WRF model has high sensitivity to the convective parameterisation scheme used
- and this prompted the present study to a full year check of the the model performance of the selected set-up. 10

**3.4 WRF-GNSS comparison: precipitation efficiency in 2013**

Figure 6 showns the monthly mean precipitation efficiency computed with GNSS and WRF for Burgas and Lovech. The very good agreement between the model and GNSS is apparent at both locations. At Burgas (Fig. 6a) the PE has minimum in August less than 1 % and maximum in May 14 %. For Lovech (Fig. 6b) the maximum PE is in May but is slightly smaller than in Burgas (12%) and the minimum is in September of about 1%. The PE at the two stations also shows differences with 15 are expected as the two stations are located in different climatic regions in Bulgaria. While Burgas is in south-east Bulgaria close to the Black Sea, Lovech is in north-west Bulgaria. The atmospheric circulation in Bulgaria is dependent on the Balkan mountains in middle of the country i.e. south of the range the Mediterranean cyclones are the main source of precipitation, while the north of the range their influence is largely reduced. The annual PE in both stations is in the range of 5.5-5.8 % from GNSS and 5.9-6.0 % from WRF, which is in agreement with the range of 5-10 % for the region found in Tuller (1971).

**Conclusions 4**

20

25

In this work GNSS tropospheric products (ZTD) with temporal resolution 5 min - are derived using the PPP processing for one year period. In order to take advantage of the high temporal resolution of GNSS products for derivation of IWV the surface pressure and temperature from the NWP WRF model is used. The WRF surface pressure and temperature was evaluated against surface observations from three synoptic stations in Bulgaria. The mean difference for surface pressure between the two datasets is less than 0.5 hPa and the correlation is over 0.989. For the temperature the largest mean difference is  $1.1^{\circ}$  C and the

correlation coefficient is over 0.957. The IWV computed with this two data-sets has a mean difference is in range of 0.1-1.1 mm.

The evaluation of WRF on annual bases shows IWV underestimation between 0.5 and 1.5 mm at five stations and overestimation at Varna and Rozhen. Varna and Rozhen have also much smaller correlation 0.9 and 0.76. The study of the monthly 30 IWV variation shows that at those locations the GNSS IWV has unexpected drop in April and March, respectfully. The reason for this drop is likely problems with station raw data. At the remaining 5 stations a very good agreement between the GNSS and WRF is observed with highest correlation in the cold part of the year i.e. March, October and December (over 0.95) and lowest correlation during the warm part of the year i.e. April to August (below 0.9). The diurnal-

The diurnal IWV cycle is investigated for Bulgaria for 2013. The diurnal variations of atmospheric water vapour affect long wave radiation, absorption of solar radiation and is related to processes such as atmospheric stability, diurnal variation of moist

- 5 convection and precipitation, surface wind convergence and evapotranspiration. Thus it is important to evaluate the IWV cycle of the WRF model shows for Bulgaria. At all stations the models has a dry bias in the range 0.5-1.5 mm. Between 00 and 01 UTC the GNSS IWV tends to be underestimated IWV, which is likely due to the processing time window used. Studies with other models show a link between IWV underestimation and overestimation of light precipitation. Such study could not be performed for Bulgaria as the precipitation observations are only available as accumulated 6 hourly values.
- In order to link the IWV and precipitation the precipitation efficiency coefficient is computed at two stations. Precipitation efficiency gives the percentage of IWV converted in precipitation. The precipitation efficiency from GNSS and WRF show very good agreement on monthly bases with a maximum in May-June and minimum in August-September. The annual precipitation efficiency in 2013 at Lovech and Burgas is about 6 %, which is within the typical values range for low elevation stations in moderate and continental climates. It will be interesting to investigate the precipitation efficiency at the mountainous stations.
- 15 but co-location of GNSS and reliable surface observations is a limiting factor for such analysis.

[revised manuscript text omitted]